

# Phylogeography of the Mesa Silverside fish *Chirostoma jordani* (Woolman, 1894) throughout the Mexican Plateau

Isai Betancourt-Resendes[1,2], Rodolfo Pérez-Rodríguez[3], Kyle R. Piller[4] and Omar Domínguez-Domínguez[3]

[1] CONAHCYT-Laboratorio de Ecología y Diversidad Faunística, Facultad de Ciencias Naturales, Universidad Autónoma de Querétaro, Querétaro, Querétaro, Mexico
[2] PIDCB, Universidad Michoacana de San Nicolás de Hidalgo, Morelia, Michoacan, Mexico
[3] Laboratorio de Biología Acuática, Facultad de Biología, Universidad Michoacana de San Nicolás de Hidalgo, Morelia, Michoacán, Mexico
[4] Department of Biological Sciences, Southeastern Louisiana University, Hammond, LA, United States of America

Corresponding author
Omar Domínguez-Domínguez,
omar.dominguez@umich.mx

## ABSTRACT

**Background**. Understanding the processes that influence distribution of organisms is a major goal in evolutionary biology. Speciation in freshwater fishes is mainly associated with the ''island-like'' model of evolution, in which the formation of land barriers between different hydrographic basins interrupts gene flow and promotes isolation. Freshwater fish therefore provide an excellent model system for macro- and micro-evolutionary studies. The Mesa Silverside, *Chirostoma jordani*, is one of the most widespread freshwater fish species in the Mexican Plateau, a geologically complex physiographic region with a long history of genesis, destruction and compartmentalization of hydrographic basins that has promoted the dispersal and isolation of freshwater fishes.

**Methods**. We used mitochondrial (*Cytb* and *D-loop*) and nuclear (first intron of the ribosomal protein S7) data and used phylogeographic and coalescent based methods to elucidate the evolutionary history of *C. jordani* throughout its distributional range on the Mexican Plateau.

**Results**. The results obtained in the present study revealed that *C. jordani* consists of two main genetic groups with geographical correspondence. Clade I occur exclusively in north-western basin and shows population structure. Clade II is widely distributed across the west, central and eastern basins without population structure. The split between these two main clades was estimated at 1.4 Mya. This cladogenetic event may be associated with the allopatric process promoted by the fragmentation and compartmentalization of hydrographic basins induced by the geological and climatic history of the Mexican Plateau.

## INTRODUCTION

Understanding the processes and patterns of biodiversity distribution is a major goal in evolutionary biology (*Mouquet et al., 2012*; *Carroll et al., 2014*). High speciation rates in freshwater fishes are mainly associated with an ''island-like'' model of evolution, where the formation of land barriers between isolated aquatic basins interrupts gene flow, which promotes the accumulation of unique adaptations, and eventually the cladogenesis of isolated populations (*Bloom et al., 2013*; *Campanella et al., 2015*; *Betancourt-Resendes, Pérez-Rodríguez & Domínguez-Domínguez, 2018*). Under this scenario, freshwater fishes are regarded as a good model system in which to integrate macro- and micro-evolutionary studies, especially in areas with unstable geological or climatic histories (*Seehausen & Wagner, 2014*).

The Mexican Plateau has been characterized by high tectonic and volcanic activity since the Miocene and is still active at present. This activity is considered the main cause of the dynamic genesis, compartmentalization and destruction of hydrographic basins (*Ferrari et al., 2012*; *Gómez-tuena, Orozco-esquivel & Ferrari, 2005*). These geological processes are also the main mechanism that limits or enhances gene flow between populations and drives speciation in freshwater fishes. (*Beltrán-López et al., 2018*; *Betancourt-Resendes, Pérez-Rodríguez & Domínguez-Domínguez, 2018*; *Betancourt-Resendes et al., 2019*; *Domínguez-Domínguez et al., 2008a*; *Domínguez-Domínguez et al., 2010*; *Pérez-Rodríguez et al., 2015*; *Pérez-Rodríguez et al., 2016*; *Piller et al., 2015*).

One of the most widespread species of freshwater fishes of the Mexican Plateau is the Mesa Silverside, *Chirostoma jordani* Woolman, 1894 (Atherinopsidae) (*Barbour, 1973a*; *Miller, Minckley & Norris, 2005*). It is distributed in the upper parts of the Panuco, Cazones and Tecolutla basins and in Totolcingo Lake to the east, the Lerma and Santiago Rivers, Cuitzeo, Valley of Mexico, and Chapala basins in the central region, the upper Ameca River, Magdalena and Atotonilco-San Marcos endorheic basins to the west and the Mezquital River to the north (*Barbour, 1973a*; *Miller, Minckley & Norris, 2005*). The Mesa Silverside is one of the smallest silverside species, occurring in both lentic and lotic ecosystems. Its ecological versatility is thought to be a key factor contributing to its broad distribution. (*Barbour, 1973a*). The species shows low genetic differentiation between some geographically isolated populations (*Bloom et al., 2009*), but high phenotypic plasticity between lentic and lotic ecosystems, suggesting that specific environmental forces drive body shape differentiation between habitats, mainly in body shape and mouth position (*Foster, Bower & Piller, 2015*). Despite the morphological differences among populations of *C. jordani*, the taxonomy of the species has remained relatively stable, with three synonymies: *Atherinichthys brevis Steindachner (1894)*, described from Cuitzeo Lake, *Chirostoma mezquital* (*Meek, 1904*), described from Mezquital River in Durango, and *Poblana hidalgoi Álvarez del Villar, 1953*, described from a dam near Tula, in Hidalgo (*Barbour, 1973b*; *Miller & Smith, 1986*). The validity of *C. mezquital* is the only remaining controversy (*Miller, Minckley & Norris, 2005*).

Due to the morphological plasticity of this species, DNA sequence data has been a powerful tool for testing phylogenetic, biogeographic, phylogeographic and species

 

boundary hypotheses within the silversides (*Betancourt-Resendes, Pérez-Rodríguez & Domínguez-Domínguez, 2018*; *Betancourt-Resendes et al., 2019*; *Piñeros et al., 2022*; *Bloom et al., 2009*; *Campanella et al., 2015*; *Barriga-Sosa et al., 2005*; *Fluker, Pezold & Minton, 2011*; *Unmack, Allen & Johnson, 2013*). Previous genetic studies of widespread freshwater fishes throughout the Mexican plateau have found geographic congruence in terms of genetic structure, which has frequently been associated with the historical configuration of the river drainages, *e.g.*, in the *Chirostoma humboldtianum* species group (*Betancourt-Resendes et al., 2019*; *Piñeros et al., 2022*), *Poeciliopsis infans* (*Beltrán-López et al., 2018*), *Goodea atripinnis* (*Beltrán-López et al., 2021*) and *Moxostoma austrinum* (*Pérez-Rodríguez et al., 2016*). Some of the populations that were identified as genetically differentiated, were later split into independent taxonomic entities, *e.g.*, *Zoogoneticus quitzeoensis –Z. purhepechus* (*Domínguez-Domínguez et al., 2008a*; *Domínguez-Domínguez et al., 2008b*), *Xenotoca eiseni –X. doadrioi –X. lyonsi* (*Piller et al., 2015*; *Domínguez-Domínguez, Bernal-Zuñiga & Piller, 2016*), *Yuriria alta –Y. amatlana* (*Domínguez-Domínguez, Pompa-Domínguez & Doadrio, 2007a*; *Domínguez-Domínguez, Pompa-Domínguez & Doadrio, 2007b*), *Notropis calientis –N. grandis –N. marhabatiensis* (*Domínguez-Domínguez et al., 2009*) and *Algansea tincella –A. amecae* (*Pérez-Rodríguez et al., 2009a*; *Pérez-Rodríguez et al., 2009b*).

Therefore, we hypothesized that geographically isolated populations of the Mesa Silverside will follow the same pattern of population and phylogenetic structure as other freshwater fishes of the Mexican Plateau (*Beltrán-López et al., 2018*; *Domínguez-Domínguez et al., 2008a*; *Domínguez-Domínguez et al., 2008b*; *Pérez-Rodríguez et al., 2016*). We conducted a phylogeographic study of *C. jordani* throughout its distribution range (*Barbour, 1973a*; *Miller, Minckley & Norris, 2005*) using a multi-locus approach, employing both mitochondrial and nuclear data sets to: (1) elucidate the evolutionary history of the group and its relationship to geological activity over time, (2) test the genetic structure of the currently isolated populations, and (3) determine the taxonomic inaccuracies within a molecular framework.

## MATERIALS & METHODS

### Sample collection and molecular data

We obtained 170 specimens of *C. jordani* from 33 localities across the species' distributional range (Table 1, Fig. 1), including the only two known northern populations: one from the Bolaños River and the other from the upper Mezquital. We captured fishes with the permission of local authorities, using electrofishing and seine netting methods, and euthanized them with tricaine-mesylate (MS-222). In addition, some specimens were obtained by local fishermen. The care and use of the animals complied with animal welfare laws, guidelines and policies, as approved by SEMARNAT-SGA/DGVS/2009/19, SEMACCDET-OS-0084/2019 and PPF/DGOPA-014/20. We preserved fin clips tissue taken from each specimen in 96% ethanol. The whole fish were fixed in 5% formalin, preserved in 70% ethanol, and identified following *Barbour (1973b)* and *Miller, Minckley & Norris (2005)*. Tissue samples and specimens were deposited in the Ichthyological Collection of the Universidad Michoacana de San Nicolas de Hidalgo (CPUM).

**Table 1  Samples localities, biogeographic regions, sequence information, and population assignment.**

| | Locality | Biogeographic region/Coordinates (UTM)/altitude(msnm) | *Cytb* sequences | *dloop* sequences | S7 sequences | Population assigment |
|---|---|---|---|---|---|---|
| **1** | La Vega | Ameca/ 619800.499-2287120.82 $N-W$/1258 | 4 | 5 | ^2 | III |
| **2** | El Tesorero | Bolaños-Santiago/709002.563 − 2526197.09 $N-W$/2121 | 2 | 3 | ^4 | I |
| **3** | Tejocotal | Cazones/589749.097 − 2226765.78 $N-W$/2127 | 5 | 9 | 6 | II |
| **4** | Cajititlan | Santiago/ 674261.47-2258395.41 $N-W$/1554 | 3 | 5 | ^10 | I |
| **5** | Petatan | Chapala/ 722833.606-2230914.75 $N-W$/1523 | 7 | 7 | 7 | I |
| **6** | Los Negritos | Chapala/ 750047.08-2220053.79 $N-W$/1520 | 13 | 5 | 0 | I |
| **7** | San Juanico | Cotija/ 741703.912-2196193.06 $N-W$/1839 | 6 | 5 | ^10 | I |
| **8** | Andocutiin | Cuitzeo/ 1305189.933-2206884.41 $N-W$/1835 | 6 | 5 | ^10 | I |
| **9** | Balneario Huingo | Cuitzeo/ 308083.243-2202811.12 $N-W$/1842 | 2 | 5 | 3 | I |
| **10** | Sengio | Cuitzeo/347332.761 − 2186273.65 $N-W$/2383 | 5 | 4 | 1 | I |
| **11** | Orandino | Lower Lerma/ 779796.889-2208878.84 $N-W$/1570 | 6 | 10 | 7 | II |
| **12** | Presa de Garabato | Santiago/ 739804.272-2282737.64 $N-W$/1714 | 2 | 2 | 0 | I |
| **13** | Magdalena | Magdalena/ 705892.034-2312789.4 $N-W$/1364 | 4 | 7 | ^8 | I |
| **14** | Guadalupe Aguilera | Mezquital/528970.567 − 2704378.12 $N-W$/1996 | 5 | 8 | ^10 | VI |
| **15** | Presa Angamacutiro | Middle Lerma/ 215027.129-2228144.01 $N-W$/1702 | 6 | 4 | 0 | I |
| **16** | Arroyo Neutla | Middle Lerma/ 306311.287-2289428.77 $N-W$/1885 | 1 | 1 | ^2 | I |
| **17** | Presa Echevereste | Middle Lerma/ 224023.558-2344597.77 $N-W$/1897 | 10 | 10 | 6 | II |
| **18** | Taretan | Middle Lerma/ 257244.148-2298541.97 $N-W$/1746 | 1 | 2 | 2 | I |
| **19** | San Francisco del Rincon | Middle Lerma/ 204225.233-2331772.92 $N-W$/1781 | 0 | 1 | 0 | I |
| **20** | Yuriria | Middle Lerma/ 280552.445-2240911.7 $N-W$/1739 | 4 | 7 | 2 | I |
| **21** | Presa de Guapango | Panuco/ 425748.229-2208809.93 $N-W$/2621 | 4 | 4 | 4 | I |
| **22** | Tepeji del Rio | Panuco/ 467199.404-2207128.25 ''N-W/2119 | 3 | 4 | ^4 | I |
| **23** | Belem del Refugio | Verde-Santiago/ 765619.211-2383092.54 $N-W$/1720 | 1 | 1 | ^2 | III |

**Table 1** (*continued*)

| | Locality | Biogeographic region/Coordinates (UTM)/altitude(msnm) | *Cytb* sequences | *dloop* sequences | S7 sequences | Population assigment |
|---|---|---|---|---|---|---|
| 24 | Lagos de Moreno | Verde-Santiago/ 191450.178-2363469.41 $N-W$/1854 | 6 | 6 | 7 | V |
| 25 | Nochistlan | Verde-Santiago/ 734535.703-2369224.73 $N-W$/1904 | 6 | 6 | ^6 | III |
| 26 | La Paz Dam | Verde-Santiago/ 230865.787-2418623.33 N-W/ | 2 | 2 | 0 | II |
| 27 | Chichimeco Dam | Verde-Santiago/ 771310.885-2436011.25 $N-W$/1953 | 5 | 8 | 5 | II |
| 28 | Ojuelos | Verde-Santiago/ 206880.187-2409825.03 $N-W$/2077 | 6 | 6 | 3 | II |
| 29 | Rio Chilerillo | Verde-Santiago/ 7637116.407-2400295.99 $N-W$/1766 | 5 | 9 | 9 | II |
| 30 | Rio Verde-Balneario Las flores | Verde-Santiago/ 726741.722-2324472.96 $N-W$/1504 | 5 | 9 | ^10 | IV |
| 31 | Rio Verde-Sn Nicolas de las Flores | Verde-Santiago/ 754349.293-2357029.14 $N-W$/1668 | 4 | 6 | ^6 | I |
| 32 | San Isidro | Verde-Santiago/794547.619−2323207.65 $N-W$/2091 | 0 | 2 | 0 | I |
| 33 | Cuemanco | Mexico Valley/ 490185.98-2134495.23 "N-W/2238 | 7 | 2 | 0 | I |
| | **Total** | | **145** | **170** | **146** | |

**Notes.**
^samples included both alleles from sequences of the first inron of the ribosomal protein S7.

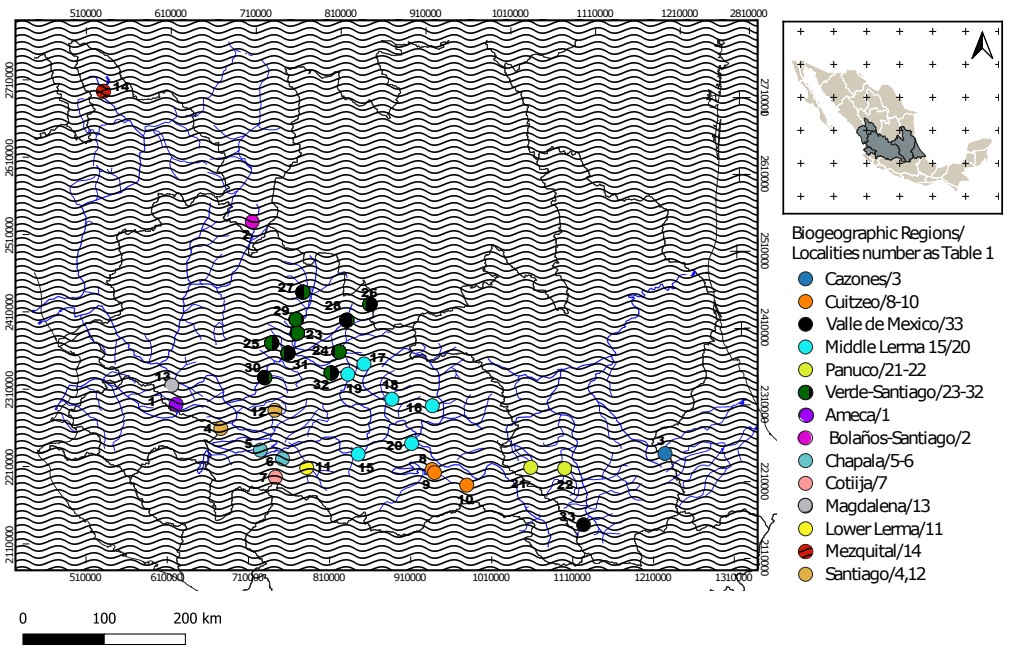

**Figure 1** Geographic locations of *Chirostoma jordani* throughout the Mexican Plateau. Circles are colored according to biogeographic regions proposed by *Dominguez-Dominguez, Doadrio & Perez-Ponce de Leon (2006)*. The numbers in the legends correspond to localities as described in Table 1.

We isolated genomic DNA using the conventional proteinase K/phenol/chloroform protocol (*Hillis, Moritz & Mable, 1996*). Polymerase chain reaction (PCR) was used to amplify the mitochondrial *Cytochrome b* locus (*Cytb*: 1,140 bp) using the primers Glud-G (*Palumbi, 1996*) and H16460 (*Perdices et al., 2002*). A fragment of the *hypervariable control region* (*D-loop*: 350 bp) also was amplified using the primers RCA and RCE (*Anderson et al., 1981*; *Kim et al., 1999*) and, a fragment of the first intron of the ribosomal protein S7 (S7: 629 bp) also was amplified using the primers S7RPEX1F and S7RPEX1R (*Chow & Hazama, 1998*). The PCR cycling conditions and primers sequences are described in Table S1. The PCR products were visualized on a 1.5% agarose gel and amplicons were submitted to MACROGEN Korea for sequencing.

The chromatograms of the recovered sequences were manually examined to eliminate potential sequencing errors. We performed the alignments manually using the software Geneious v.4.8.5. We translated the *Cytb* alignments to amino acids to verify the absence of stop codons along the sequence. We resolved the heterozygous sites for the S7 sequences using the algorithm provided by PHASE 2.0 (*Stephens, Smith & Donnelly, 2001*), as implemented in DnaSP *v.* 5.10 (*Librado & Rozas, 2009*), employing the default parameters. The non-recombination test of S7 was conducted using the software DnaSP *v.* 5.10 (*Librado & Rozas, 2009*). Additionally, we also gathered sequences data from several related species including *Chirostoma estor*, *Chirostoma humboldtianum*, *Chirostoma chapalae*, *Chirostoma sphyraena*, and *Chirostoma attenuatum*. These species were used as outgroups in the gene tree analyses. The DNA sequences were deposited in NCBI databases (https://www.ncbi.nlm.nih.gov/), accession numbers are included in Table S2. For the phylogenetic analyses, the best-fit model for each marker was obtained in PartitionFinder v.1.1.0 (*Lanfear et al., 2012*) using the Akaike Information Criterion (AIC).

## Gene trees, genetic distance and haplotype networks

We inferred gene trees of haplotype data files of both mtDNA (*Cytb* and *D-loop*) and nDNA (S7) using both, maximum likelihood (ML) and Bayesian Inference (BI) approaches. The ML was run in RAxML (*Stamatakis, Hoover & Rougemont, 2008*; *Silvestro & Michalak, 2012*; *Stamatakis, 2014*) with the AICc-select model with three gene partitions. Bayesian analyses were run in MrBayes 3.2.2 (*Ronquist et al., 2012*). Two independent runs were implemented with four MCMCs and 10,000,000 generations, sampling every 100 trees. Chain convergence was verified by a suitable effective sample size (ESSs >200) for all parameters in Tracer package *v.* 1.7, discarding 10% of generations as burn-in (*Rambaut et al., 2018*; *Sahlin, 2011*). Additionally, we estimated the uncorrelated *p* genetic distance based on both mtDNA and nDNA using "APE" v.5.8 (*Paradis, Claude & Strimmer, 2004*) package implemented in R.

To examine the geographic distribution of haplotypes, we constructed a haplotype network for each independent locus in HaploView *v.* 4.2 (*Barrett et al., 2005*), based on an unrooted ML tree obtained in RAxML, through the CIPRES Science Gateway v.3.3 (*Miller et al., 2015*).

## Genetic structure

To determine the amount of genetic variation partitioned within and among populations, an analysis of molecular variance (AMOVA) was performed in ARLEQUIN (*Excoffier, Smouse & Quattro, 1992*; *Excoffier & Lischer, 2010*) for the three separate loci. This analysis was run with significance levels set at $\alpha = 0.05$ and 10,000 random permutations.

We used Hierarchical Bayesian Analysis of Populations Structure (hBAPS) (*Cheng et al., 2013*), to examine the genetic population structure, using the Single Nucleotide Polymorphism (SNPs) matrix from the mtDNA and nDNA sequences. The analysis was carried out in the RhierBAPS (rBAPS) package implemented in R language (*Cheng et al., 2013*; *Hill et al., 2019*). The initial number of clusters assigned was equal to the number of populations with a length of run = 1,000,000. To plot the groups obtained, we used a guide tree result estimated using a pml: Likelihood in package "phangorn" (*Schliep, 2011*) implemented in R. The bar-plot graphics of probability assignment of rBAPS results were generated in R package. Additionally, we calculated the Φct, Φst and Φsc values in ARLEQUIN (*Excoffier, Smouse & Quattro, 1992*; *Excoffier & Lischer, 2010*) for the main clusters recovered in hBASP to mtDNA and nDNA.

## Time calibration

To estimate the divergence time within *C. jordani* we concatenated mtDNA and nDNA haplotypes into a single data set. The time calibration analysis was performed using BEAST 2 package (*Bouckaert et al., 2014*), using a coalescent prior (Bayesian skyline plot). We included samples of related species as described in the gene tree analysis. We used an uncorrelated lognormal clock. Due to a limited fossil record for the group, the molecular clock was calibrated with *Cytb* rate of 0.005 substitutions/site/Mya estimated for Atheriniformes fishes (*Campanella et al., 2015*; *Unmack, Allen & Johnson, 2013*). We estimated the mutation rate of *D-loop* and S7 relative to *Cytb*, using a normal prior (0.005 ± 0.001). The best-fit models of nucleotide evolution were used as estimated in PartitionFinder. The assays were performed in BEAST 2 implemented on the web server CIPRES Science Gateway *v*. 3.3 (*Miller et al., 2015*). The analysis was run for 100,000,000 generations sampling each 1,000 generations. Ten percent of the runs were discarded as burn-in. Two independent runs were performed and both files were combined using LogCombiner v.2 (*Bouckaert et al., 2014*). Posterior probability density of the combined tree file was summarized with TreeAnnotator *v*. 2 (*Helfrich et al., 2018*).

## RESULTS

### Molecular data

One hundred seventy specimens were collected from 33 localities from 14 biogeographic regions (*sensu Domínguez-Domínguez et al., 2010*) in the Mexican Plateau (Fig. 1; Table 1). One hundred forty-five sequences were obtained for the mitochondrial *Cytb* locus of 1,000 bp of length. One hundred seventy sequences were obtained for the mitochondrial *D-loop* locus with 333 bp. For the S7 nuclear locus, 99 sequences were obtained with 629 bp, and all heterozygous sites found in the S7 sequences were successfully resolved for SNP variation (phase threshold value >85%), producing 146 genotypes (Table 1). The best-fit

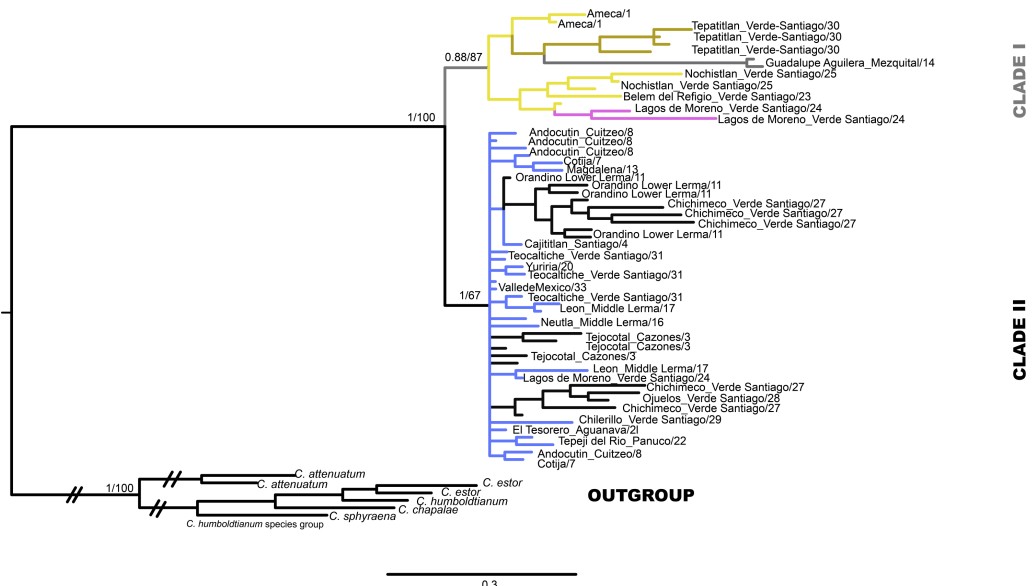

**Figure 2** Gene tree based in haplotype data set of concatenated *Cytb* and *D-loop* mitochondrial DNA sequences, using both Bayesian and maximum likelihood approaches. Numbers on branches represent the posterior probability (0-1) for Bayesian estimation and bootstrap support (0-100) for maximum likelihood estimation. Color branches are according to main genetic clusters recovered in the BAPS analysis. The numbers in the labels correspond to localities described in Table 1 and Fig. 1.

models of nucleotide evolution were as follows, for *Cytb* was TrN + G, GTR +I +G for *D-loop* and GTR+G for the S7.

## Gene trees, genetic distance and haplotype networks

The mtDNA gene tree using BI recovered two clades (Fig. 2). Clade I (Fig. 1) is not supported by BI but is supported by ML and includes haplotypes from La Vega (Ameca basin), Belem del Refugio, Nochistlan, Lagos de Moreno, and Tepatitlan in the Verde River (Verde-Santiago Basin), as well as the Guadalupe Victoria dam (Mezquital basin). Clade II (Fig. 1) is well supported by BI (posterior probabilities (pp) = 1) but not by ML, and includes haplotypes from Cazones, Chapala, Cotija, Cuitzeo, Bolaños-Santiago, Magdalena, Lerma, Pánuco, Valle de México, El Chichimeco, La Paz, Ojuelos, Río el Chilerillo, and San Nicolas de las Flores (Verde-Santiago basins) (Fig. 2). The ML estimation also recovered two clades, with the same population distribution as in the BI analysis. Clade I is supported with a bootstrap of 87, but Clade II is not supported by ML (Fig. 2). The S7 haplotype data gene tree recovered a basal polytomy among populations without geographical clustering in both BI and ML approaches (Fig. S1). The uncorrected P-distance between Clade I and Clade II was estimated at 3% using mtDNA sequences (Fig. S2) and 1% with nDNA (Fig. S3).

The haplotype network analysis of the mitochondrial locus reveals the presence of two distinct haplogroups. Within Haplogroup 1, internal differentiation is observed, with four geographically distinct groups: two distributed in different tributaries of the Verde-Santiago drainage, separated by four to seven mutation steps from the nearest haplotype; one in

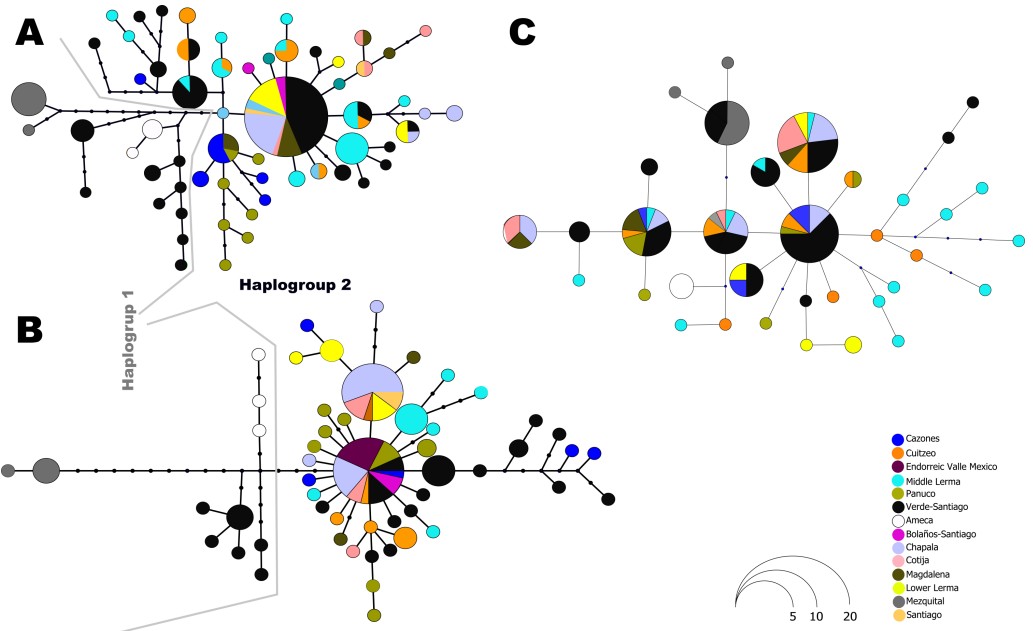

**Figure 3** Haplotype network for all independent loci: (A) *D-loop hypervariable control region,* (B) *Cytochrome b* and (C) *First intron of the Ribosomal Protein S7.* The size of the circles is proportional to the frequency of the haplotype according to the figure. Colors in circles represent the locality or regions as shown in the legends.

the Ameca drainage, separated by three to five mutation steps; and one in the Mezquital drainage, separated by ten to eleven mutation steps (Figs. 1 and 2A–2B). Haplogroup 2 clusters the remaining samples from western, central, and eastern drainages, including the Verde River (within the Santiago drainage), as well as samples ranging from Magdalena in the west (number 13 in Fig. 1) to Cazones in the east (number 3 in Fig. 1). The S7 network does not show distinct haplogroup formation, although the Mezquital and Ameca samples display genetic distinctiveness, with some mixing observed between the Verde and Mezquital populations. The remaining locations share common haplotypes (Fig. 3C).

## Genetic structure

The AMOVA analysis of the mtDNA, without *a priori* groupings, showed a high and significative $\Phi$st value ($\Phi$st = 0.51), rejecting the null hypothesis of panmixia. The majority of the total variation (50.61%) was explained by differences between populations ($\Phi$st>$\Phi$sc). In contrast, for the nDNA sequences, the majority of the variation (67.1%) was explained within populations, with only 35.52% explained between populations ($\Phi$st<$\Phi$sc). The $\Phi$st value differed significantly from 0 ($\Phi$st = 0.329), again rejecting the null hypothesis (Table 2).

The hBAPS analysis for mtDNA partitioned the genetic variation in six clusters. Genetic cluster (K1) included samples from Cuitzeo, Bolaños, Cotija, Magdalena, Middle Lerma, Chapala, Santiago, Verde Santiago, Panuco and Valle de México. The second genetic cluster (K2) grouped samples from Cazones, Lower Lerma, and Verde Santiago. The third

**Table 2** Analysis of molecular variance (AMOVA) without *a priori* grouping, based on both mtDNA and nDNA sequences.

| Source of variations | d.f | Sum of squares | Variance components | Percent of variation |
|---|---|---|---|---|
| **mtDNA** | | | | |
| Among populations | 25 | 408.58 | 4.30590 Va | 50.61 |
| Within populations | 48 | 201.73 | 4.20278 Vb | 49.39 |
| Total | 73 | 610.31 | 8.51 | |
| Fixation Index | ($\Phi$st) = 0.51 | | | |
| **ntDNA** | | | | |
| Among populations | 25 | 81.27 | 0.41708 Va | 32.9 |
| Within populations | 125 | 106.33 | 0.85065 Vb | 67.1 |
| Total | 150 | 187.6 | 1.27 | |
| Fixation Index | ($\Phi$st) = 0.329 | | | |

genetic cluster (K3) grouped samples from the Ameca and, Verde Santiago. The fourth (K4) and fifth (K5) genetic clusters grouped samples from Verde Santiago. Finally, the sixth genetic cluster (K6) grouped samples from Mezquital (Figs. 2 and 4). The F-statistics for this arrangement were: $\Phi$ct = 0.36, $\Phi$st = 0.55 and $\Phi$sc = 0.29 ($\Phi$ct>$\Phi$sc). For nDNA, we recovered three genetic clusters, but with extensive admixture between geographical locations. The first cluster (K1) grouped samples from Chapala, Cuitzeo, Middle and Low Lerma, Magdalena and Verde Santiago Basin. The second genetic cluster (K2) grouped samples from Ameca, Bolaños-Santiago, Cazones, Chapala, Cuitzeo, Cotija, Lower Lerma, Magdalena, Middle Lerma, Verde Santiago and Panuco basin, and the third genetic cluster (K3) grouped samples of Mezquital and Verde Santiago (Nochistlan) basin (Fig. 5). The F-statistics for this arrangement were: $\Phi$ct = 0.17, $\Phi$st = 0.38 and $\Phi$sc = 0.25.

### Time calibration analysis

The divergence time analysis showed the split between clade I and clade II dated in the Pleistocene *ca.* 1.4 Mya (95% HDP 0.87−2.1 Mya) (Fig. 6).

## DISCUSSION

The active geological and climatic history of the Mexican Plateau has influenced the dynamism of genesis, destruction, and compartmentalization of hydrographic systems, and is the main factor that promotes both, the cladogenesis and extension of the distributional ranges of freshwater fishes in this region (*Betancourt-Resendes, Pérez-Rodríguez & Domínguez-Domínguez, 2018*; *Betancourt-Resendes et al., 2019*; *Doadrio & Domínguez, 2004*; *Dominguez-Dominguez, Doadrio & Perez-Ponce de Leon, 2006*; *Domínguez-Domínguez et al., 2008a*; *Pérez-Rodríguez et al., 2009a*; *Pérez-Rodríguez et al., 2009b*; *Piller et al., 2015*; *Beltrán-López et al., 2018*; *Beltrán-López et al., 2021*). Previous studies of species *Chirostoma*, with narrower distributional ranges, have identified allopatry as the main force shaping the evolution of this group (*Betancourt-Resendes, Pérez-Rodríguez & Domínguez-Domínguez, 2018*; *Betancourt-Resendes et al., 2019*; *Piñeros et al., 2022*).

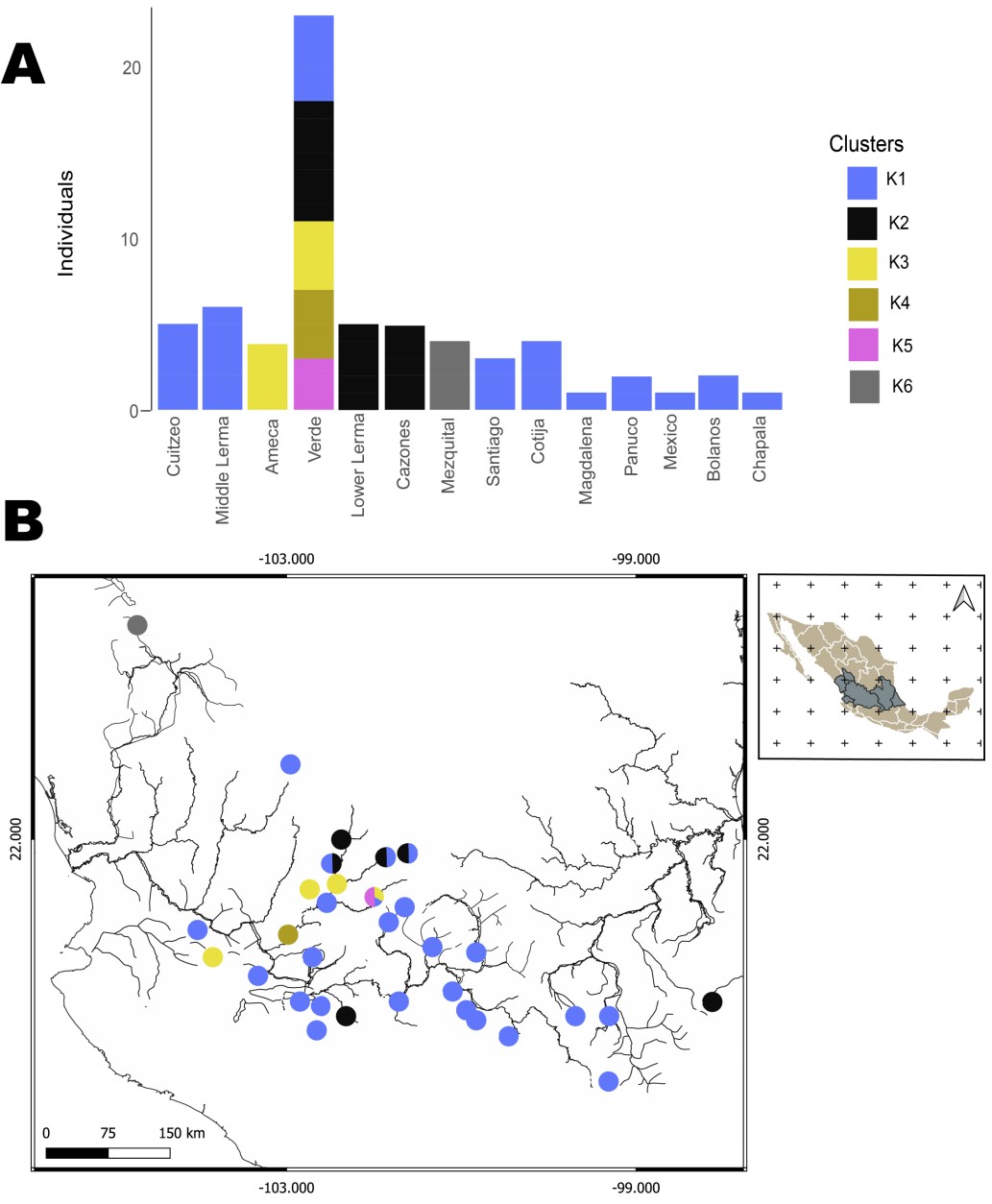

**Figure 4** **Hierarchical Bayesian Analysis of Populations Structure (hBAPS) using the Single Nucleotide Polymorphism (SNPs) matrix from the mtDNA sequences.** (A) represents the major genetic clusters recovered with hBAPS using mtDNA (*Cytb, Dloop*) SNPs, (B) represents the main clusters plotted on a map according to geographic distribution.

## Split of the two main clades

Our results support the existence of two main clades within *C. jordani* (Fig. 2). The split was dated *ca* 1.4 Mya (Fig. 6). Clade I is confined to western (Ameca, Verde-Santiago) and northern basins (Mezquital), while clade II is widely distributed in western, central, and eastern basins on the Mexican Plateau (Figs. 1 and 2), showing mixtures of haplotypes

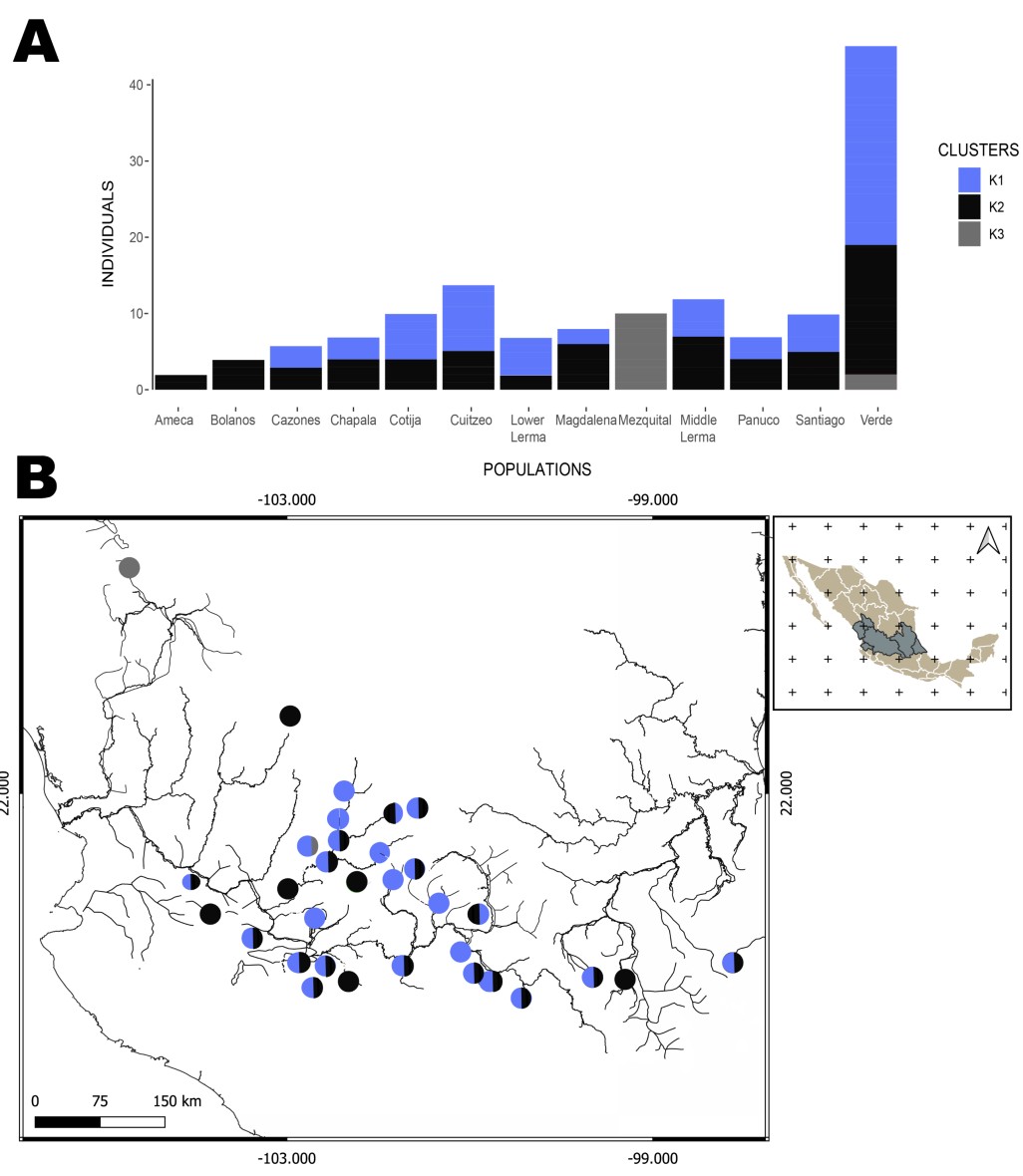

**Figure 5** **Hierarchical Bayesian Analysis of Populations Structure (hBAPS) using the Single Nucleotide Polymorphism (SNPs) matrix from the mtDNA and nDNA sequences.** (A) Represents the major genetic clusters recovered with hBAPS using a nDNA SNPs, (B) The main clusters plotted on a map according to geographic distribution.

between regions, except for the Valle de Mexico, Panuco and Cazones samples for *D-loop* (Fig. 3A). The biogeographic scenario that explains the cladogenetic events in *C. jordani* is complex, mainly because the mixture of populations belonging to the two haplogroups in the Verde River. This area has been geologically active in the last 3 Mya. We hypothesize that the isolation between Clade I and Clade II and the mixture of populations belonging to different haplogroups, has been promoted by the high tectonic and volcanic activity in the "triple junction area" between 3 and 1 Mya, which allowed for the isolation

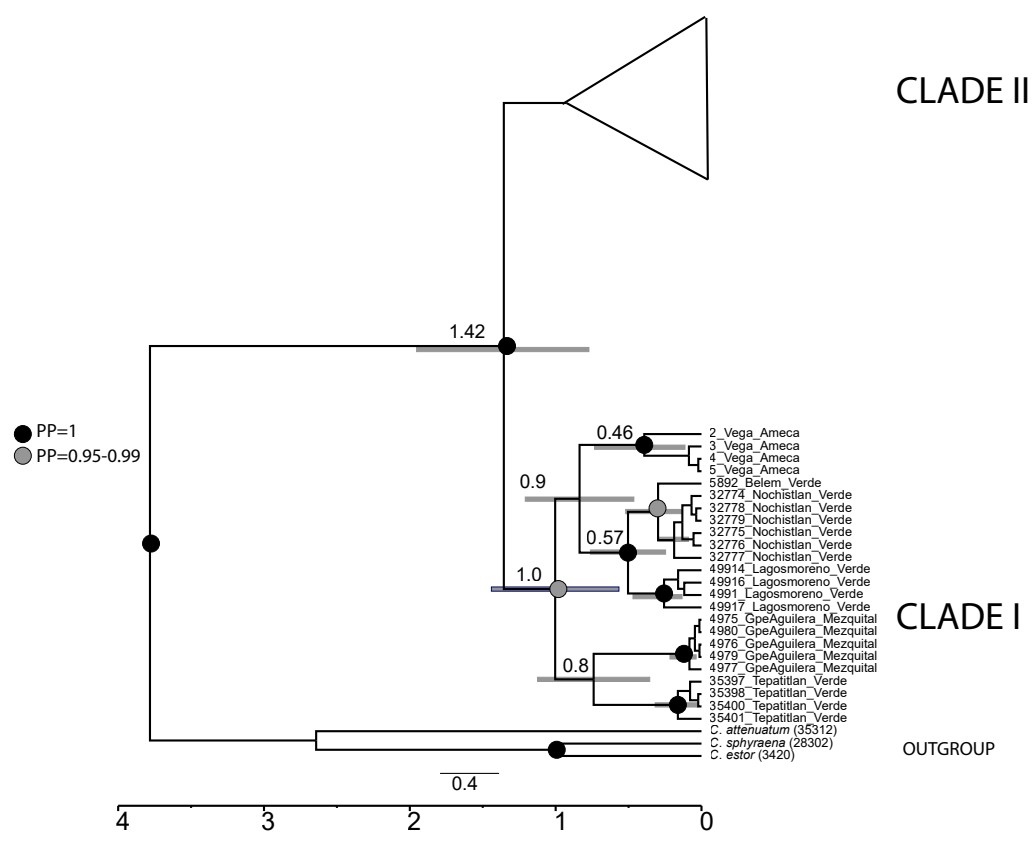

**Figure 6** **Calibrated analysis of *C. jordani* populations based on three loci (*D-loop*, *Cytb* and S7).** The circles at the nodes represent the posterior probability from the Bayesian analysis, black circles = 1, grey circles = 0.95. no circles < 0.95. Grey lines in nodes represented the 95% HDP.

and reconnection of the upper Ameca, Verde, Lerma Rivers and Chapala Lake regions (*Rosas-Elguera & Urrutia-Fucugauchi, 1998*; *Nieto-Samaniego et al., 1999*). The influence of the geological activity of the "triple junction" in the evolution patterns of the Mexican ichthyofauna has been extensively discussed by other authors (*Barbour, 1973b*; *Smith, Cavender & Miller, 1975*; *Miller, Minckley & Norris, 2005*; *Domínguez-Domínguez et al., 2010*; *Pérez-Rodríguez et al., 2016*; *Beltrán-López et al., 2018*; *Beltrán-López et al., 2021*). The climatic instability during the early Pleistocene, a period characterized by intercalated humid and dry periods, has been thought to have had an important influence on the evolutionary and demographic history of the Mexican ichthyofauna (*Barbour, 1973b*; *Mateos, Sanjur & Vrijenhoek, 2002*; *Domínguez-Domínguez et al., 2010*; *Pérez-Rodríguez et al., 2009a*; *Pérez-Rodríguez et al., 2009b*; *Pérez-Rodríguez et al., 2015*; *Beltrán-López et al., 2018*; *Betancourt-Resendes, Pérez-Rodríguez & Domínguez-Domínguez, 2018*; *Betancourt-Resendes et al., 2019*; *García-Martínez et al., 2015*; *García-Martínez et al., 2020*). The climatic scenario may have influenced the interchange of fauna in the tributaries of the Verde and Lerma Rivers, promoting the connection of headwater tributaries in wet periods, and the mixture of the Verde population in both haplogroups,

## Phylogeographic patterns of clade I

Our phylogeographic and population genetic results support well-structured genetic groups with geographic correspondence within Clade I/Haplogroup 1 based on mtDNA and partially for nDNA (Figs. 2, 3, 4 and 5). The TRMCA was dated *ca.* 1 Mya (Fig. 6). Within this clade, one group occurred in the north (Mezquital Basin in Mesa del Norte) and three in the west (Ameca and Verde-Santiago basins in central Mexico).

The biogeographic relationship between the Mezquital Basin and the central basins of Mexico has been widely recognized. The genus *Characodon* (Goodeidae) and the species *Moxostoma milleri* (Catostomidae), are distributed in the Mezquital River basin and have their closest relatives distributed in central Mexico (*Doadrio & Domínguez, 2004*; *Domínguez-Domínguez et al., 2010*; *Pérez-Rodríguez et al., 2015*). However, the phylogenetic relationships between these groups show older divergence times than *Chirostoma* (<1Mya), being *ca.* 4.5 Mya for the split of *M. milleri*, and *ca.* 15.5 Mya for the split of *Characodon*. Although is not clear how the Mezquital River was connected to the central Mexican basins in recent geological times, according to the peripheral position of the Mezquital samples in the three haplotype networks, a plausible isolation scenario could be the range extension of the MRCA from the Verde-Santiago drainage to the Mezquital River, with a posterior vicariant event that generated the allopatric pattern. The likely connection between the Mezquital River and central Mexico could be through the Guadiana Valley, a high plain in the upper Mezquital that border in its southern portion with the rivers that drain to the Santiago River. The accumulation of Pleistocene alluvial deposits within the Guadiana Valley, in synergy with the volcanic activity (*Albritton, 1958*; *Aranda-Gómez et al., 2018*) and the climatic changes that occurred during Pleistocene, could have caused the isolation of the MRCA of *C. jordani* from Mezquital drainage.

The taxonomic status of *C. jordani* from the Mezquital drainage has been highly discussed. In morphological analyses, the Mezquital samples show extensive morphological polymorphism when compared with specimens of *C. jordani* collected from throughout the entire distribution of the species, and has therefore, largely been considered a synonym of *C. jordani* (*Barbour, 1973a*). Even in the Integrated Taxonomic Information System-ITIS (Last accessed 10/24/2023, http://www.itis.gov) and FishBase (Last accessed 10/24/2023, https://www.fishbase.se/summary/Chirostoma-jordani.html), it appears as a synonym of *C. jordani*. However, *Miller, Minckley & Norris (2005)* recognized *C. mezquital* as a valid species based on characters associated with body height measurements. The results presented herein support the genetic distinctiveness of the Mezquital samples, forming an independent group in haplotype network, and hBAPS analyses (Figs. 2–5). However, other populations, including the Ameca and Verde River populations, also show genetic differences. According to the results presented in this study, we propose an exhaustive integrative taxonomic study to elucidate the taxonomic status of *C. jordani* population from Mezquital and other genetically differentiated populations.

The samples collected in the Verde River basin show three genetically segregated groups inhabiting this basin. Two groups were clustered into Clade I and the other group of samples was clustered in Clade II (Figs. 2–5). As previous studies mentioned, the existence of genetically differentiated populations within the Verde River could be the result of the

existence of temporally and spatially independent events of colonization in the Verde River (Figs. 2–5), as was previously proposed for *Poeciliopsis infans, Goodea atripinnis, Algansea tincella* and *Notropis calientis* (*Domínguez-Domínguez et al., 2008a*; *Domínguez-Domínguez et al., 2009*; *Pérez-Rodríguez et al., 2009b*; *Beltrán-López et al., 2018*; *Beltrán-López et al., 2021*). Hence, we hypothesize that the first colonization event to the Verde River was through the postulated connection of the Verde Paleoriver and Chapala Paleolake (*Smith, Cavender & Miller, 1975*; *Barbour, 1973a*; *Miller & Smith, 1986*; *Aranda-Gómez, Henry & Luhr, 2000*; *Miller, Minckley & Norris, 2005*). A second colonization event seems to have occurred through a river capture event between the Middle Lerma and Verde River. The location of Lagos de Moreno, Nochistlan and Belen del Refugio is geographically close to the headwaters of the Turbio River, a tributary of the Lerma River. In this case, the intermontane plains that separate both river systems could function as corridors during wet and flood periods. This hypothetical connection has been recognized previously as a dispersal route for the ancestor of the species complex belonging to the atherinopsid silversides and other species (*Barbour, 1973b*; *Bloom et al., 2012*). A third invasion of the Verde River seems to be also related to stream captures of the Lerma and Verde Rivers; these events appear to have been recent enough to prevent haplotype sorting in any of the three sampled loci (Figs. 2–5).

The isolation and cladogenesis of the samples of *Chirostoma* from the Ameca basin are not surprising, since seven endemic freshwater fishes that occurred in this river basin have their closest relatives in Central Mexico drainages, like the leuciscids *Algansea ameca, Notropis amecae* and *Yuriria amatlana,* the goodeids *Zoogoneticus tequila, Allodontichthys polylepis, Xenotoca doadrioi, Skiffia francesae* and *Allotoca goslinei*, and the catostomid *Moxostoma mascotae* (*Miller & Smith, 1986*; *López-López & Paulo-Maya, 2001*; *Doadrio & Domínguez, 2004*; *Domínguez-Domínguez, Pompa-Domínguez & Doadrio, 2007a*; *Domínguez-Domínguez, Pompa-Domínguez & Doadrio, 2007b*; *Domínguez-Domínguez et al., 2008a*; *Domínguez-Domínguez et al., 2009*; *Domínguez-Domínguez et al., 2010*; *Dominguez-Dominguez, Doadrio & Perez-Ponce de Leon, 2006*; *Piller et al., 2015*; *Pérez-Rodríguez et al., 2009a*; *Pérez-Rodríguez et al., 2009b*; *Pérez-Rodríguez et al., 2016*; *Beltrán-López et al., 2018*). The high endemicity of the Ameca River seems to be related to a long history of volcanism and tectonism at different geological times, generating the isolation, connection and compartmentalization of the Ameca River over time, which includes several surrounding drainages (*Rosas-Elguera & Urrutia-Fucugauchi, 1998*; *Garduño & Tibaldi, 1991*). Our results support a recent isolation event of the Ameca River, since we date the cladogenesis of the Ameca group at *ca.* 0.90 Mya (Fig. 6). Two recent hydrographic connections and interchange of fauna between Ameca River and the contiguous basins have been postulated with the first through the Atotonilco, San Marcos and Zacoalco-Ameca Paleolakes area (*Smith, Cavender & Miller, 1975*), with a subsequent isolation event, occurred less than *ca* 1 Mya. This isolation seems to be promoted by Pleistocene volcanism and the intense tectonic activity of the so-called triple junction (*Rosas-Elguera & Urrutia-Fucugauchi, 1998*; *Garduño & Tibaldi, 1991*), as has also been postulated for the isolation of *P. infans, Ameca splendens, Xenotoca melanosoma* and *Z. purhepechus* populations in the Ameca River (*Domínguez-Domínguez et al., 2008a*; *Beltrán-López et al.,*

*2018*). A second connection event was proposed for *P. infans* between the Ameca River and the Verde-Santiago River drainage, suggesting that these basins were connected until very recent geological time through stream capture of the Ameca and Verde Rivers, which was facilitated by the volcanism in the Tepic-Zacoalco graben (*Mateos, Sanjur & Vrijenhoek, 2002*; *Beltrán-López et al., 2018*). However, we cannot support either of these two scenarios, due to the lack of samples of *Chirostoma* from the Sayula-San Marcos-Atotonilco lakes area, despite the high sample effort conducted.

## Phylogeographic patterns of Clade II

The Clade II, shows a widely distributed group in the central and eastern drainages of central Mexico (in an area > 130,000 km$^2$), from Bolaños River in the Santiago Drainage in the Northwest, Ameca and Magdalena in the Southwest, to Cazones to the East, with a high haplotype mixture (Figs. 3 and 5). The group has been evolving in central Mexico over the last ~1 Myr. This genetic cohesiveness of widely distributed species in central Mexico is not common, since most of the co-distributed species already studied are structured in at least one of the locations where haplotypes of the *C. jordani* central group are distributed, as is the case of *Zoogoneticus quitzeoensis* (*Domínguez-Domínguez et al., 2008a*), *Notropis calientis* (*Domínguez-Domínguez et al., 2009*), *Moxostoma austrinum* (*Pérez-Rodríguez et al., 2016*), *Algansea* spp (*Pérez-Rodríguez et al., 2009a*; *Pérez-Rodríguez et al., 2009b*), and *Poeciliopsis infans* (*Mateos, Sanjur & Vrijenhoek, 2002*; *Beltrán-López et al., 2018*).

The genetic homogeneity could be related to ancient connectivity between the central basins across several periods during the Pleistocene, that have been widely discussed in goodeids (*Doadrio & Domínguez, 2004*; *Domínguez-Domínguez et al., 2010*; *Beltrán-López et al., 2021*), leuciscids (*Schönhuth, 2002*; *Domínguez-Domínguez, Pompa-Domínguez & Doadrio, 2007a*; *Domínguez-Domínguez, Pompa-Domínguez & Doadrio, 2007b*; *Pérez-Rodríguez et al., 2009a*; *Pérez-Rodríguez et al., 2009b*; *García-Andrade et al., 2021*) and poecilids (*Mateos, Sanjur & Vrijenhoek, 2002*; *Beltrán-López et al., 2018*), as well as to the dispersal capacity of the species. In fishes, this capacity is highly dependent on body shape (*Sfakiotakis, Lane & Davies, 1999*; *Triantafyllou, Triantafyllou & Yue, 2000*; *Langerhans & Reznick, 2009*), whereas the survival rate depends on plasticity and adaptability for establishment in a new habitat (*Seehausen & Wagner, 2014*). In particular, *C. jordani* exhibits a high degree of morphological variation (*Barbour, 1973b*), part of which is associated with divergent habitats, and differential selective pressures (*Foster, Bower & Piller, 2015*). Moreover, its small size and ecological versatility (*Barbour, 1973a*) gives *C. jordani* the capacity to widely disperse throughout the lakes, rivers, and streams of central Mexico, and also to survive in newly invaded environments, taking advantage of possible historical dispersal routes in currently isolated basins (*Domínguez-Domínguez et al., 2008a*; *Pérez-Rodríguez et al., 2009a*; *Pérez-Rodríguez et al., 2009b*; *Beltrán-López et al., 2018*).

## CONCLUSIONS

We observed recent genetic differentiation (<1.4 Mya) within the *C. jordani* samples analyzed in this study. The phylogeographic patterns and population genetics results revealed significant evolutionary insights. We identified a well-differentiated evolutionary unit in the Ameca and Mezquital Basins, which may represent an evolutionarily independent lineage. These findings underscore the importance of the Mezquital and Ameca Rivers basins as crucial regions for the evolution and diversification of aquatic fauna in the Mexican Plateau and the conservation of endemic and genetically unique ichthyofauna. The discovery of three genetic clusters within the Verde Drainage, two of which are genetically distinct from other basin samples, further highlights the complexity of freshwater fish evolution in the geologically and climatically dynamic Mexican Plateau.

While several studies on widespread freshwater fishes across the Mexican Plateau have shown that the area's geological and climatic history promotes species diversification, the lack of genetic differentiation among different basins for *C. jordani*, as seen in Haplogroup II, seems to be linked to the species morphological and ecological plasticity. This adaptability has enabled the species to disperse across connected water bodies and thrive in the environmental conditions of newly colonized areas.

## ACKNOWLEDGEMENTS

We thank to B. García-Andrade, G. Beltrán-Lopez and F. Mar-Silva, for their field support. We would also like to thank the CIPRES Cyberinfrastructure for Phylogenetic Research XSEDE for their computational support. We thank the anonymous reviewers for their valuable comments.

### Funding

This work was supported by Consejo Nacional de Ciencia y Tecnología (CONACyT) which granted a scholarship to Isai Betancourt-Resendes (N° de Beca 380522 y N° de Registro 254124). The División de estudios de Posgrado de la Universidad Michoacana de San Nicolás de Hidalgo funded the English editing of the article. The Consejo de la Investigación Científica, UMSNH (to Omar Domínguez-Domínguez) and the National Science Foundation (NSF 1354930 to Kyle R. Piller). The funders had no role in study design, data collection and analysis, decision to publish, or preparation of the manuscript.

### Grant Disclosures

The following grant information was disclosed by the authors:
Consejo Nacional de Ciencia y Tecnología (CONACyT):  No de Beca 380522 y No de Registro 254124.
The División de estudios de Posgrado de la Universidad Michoacana de San Nicolás de Hidalgo.
The Consejo de la Investigación Científica, UMSNH.
The National Science Foundation: NSF 1354930.

## Competing Interests

The authors declare there are no competing interests.

## Author Contributions

- Isai Betancourt-Resendes conceived and designed the experiments, performed the experiments, analyzed the data, prepared figures and/or tables, authored or reviewed drafts of the article, and approved the final draft.
- Rodolfo Pérez-Rodríguez analyzed the data, prepared figures and/or tables, authored or reviewed drafts of the article, and approved the final draft.
- Kyle R. Piller analyzed the data, authored or reviewed drafts of the article, english review, and approved the final draft.
- Omar Domínguez-Domínguez conceived and designed the experiments, analyzed the data, prepared figures and/or tables, authored or reviewed drafts of the article, and approved the final draft.

## Field Study Permissions

The following information was supplied relating to field study approvals (i.e., approving body and any reference numbers):

The care and use of the animals complied with animal welfare laws, guidelines and policies, as approved by SEMARNAT-SGA/DGVS/2009/19, SEMACCDET-OS-0084/2019 and PPF/DGOPA-014/20.

## Data Availability

The *Cythocrome b* locus sequences are available at GenBank: PP454718 to PP454722 and PP475811 to PP475939.

The *D-loop* locus sequences are available at GenBank: PP454723 to PP454730 and PP475940 to PP476095.

The first intron of ribosomal protein S7 locus sequences are available at GenBank: PP454731 to PP454734 and PP484943 to PP485034.

## Supplemental Information

Supplemental information for this article can be found online at http://dx.doi.org/10.7717/peerj.18256#supplemental-information.

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
