# Peer review of "Phylogeography of the Mesa Silverside fish Chirostoma jordani (Woolman, 1894) throughout the Mexican Plateau"

_PeerJ, doi:10.7717/peerj.18256_

## Round 0.1 · original submission · Major Revisions

We received comments of two reviewers. Most of them are fairly minor (grammar, text, style) and should be easy to address. Reviewer 2 also raises a few concerns about the interpretation of the phylogenetic analyses. I do agree with that assessment but I think this can be addressed by some changes in the discussion section. Similarly, I find the biogeographic discussions in the discussion very lengthy. Not that there is a technical limit in an online journal but I think some shortening of the text would make it more accessible to readers.

**Language Note:** The review process has identified that the English language must be improved. PeerJ can provide language editing services - please contact us at [email protected] for pricing (be sure to provide your manuscript number and title). Alternatively, you should make your own arrangements to improve the language quality and provide details in your response letter. – PeerJ Staff

Reviewer 1 ·

Basic reporting

The article is an interesting study looking at the phylogeography of the Mexican silverside, working across population genetic and phylogenetic scales through the sequencing of one nuclear and two mitochondrial genes. I did not have any major concerns with the overall aspects of the article, and I commend the authors on a well-written manuscript: the introduction has sufficient background and covers topics relevant to the study, including hypotheses for the authors' expectations; the methods are sufficiently detailed to explain what was done; and the results detail the important findings. The figures and tables present the major findings in an easily accessible manner. The discussion provides context for the findings. The raw data are available on NCBI. I had only small grammatical suggestions for the authors, which I have detailed in the section below.

Experimental design

This article details original research by the authors that provides an interesting case study of evolution of freshwater fishes on the Mexican plateau. The research questions are clear, and the methods sufficient for addressing these questions. I have no concerns about the experimental design.

Validity of the findings

The findings in this study are supported by the data and linked to the original research question. My only small concern in this area is that there seems to be a gap in spatial sampling between the Cuitzeo sample (Site 14), Bolanos-Santiago (Site 2), and the others. I did not see the authors address this spatial gap in their discussion; while I do not think it likely that this would affect their overall conclusions, I think it would be good to acknowledge that such gaps may affect the population structure results. If there is isolation-by-distance, such a gap in samples may make this look like clustering instead. Alternatively, because there are not other nearby samples in the dataset, the Cuitzeo site may appear to cluster with some of the other sites even if it is relatively distinct. For this reason, I would recommend the authors including an analysis of the relationship between geographic distance (here, it would probably make sense to use river distance) and genetic distance between individuals or populations to check to see whether this may be influencing conclusions.

In addition, I have the following specific comment for the authors:
L152: What was used as the threshold for "suitable effective sample size"? How was 10% chosen as the threshold for burn in? If there are reasons for these things, it would be good to state them or cite where they came from.

Additional comments

In addition to the above comments, I have some minor spelling/grammatical suggestions for the authors:

L32: "coalescent-based methods"
L37-39: "Clade I exclusively inhabits the north and western basins, and also shows population structure. Clade II is widespread, inhabiting the west, central, and eastern basins without population structure."
L40: "seems"
L69-73: Long sentence, would probably benefit from being broken up into two
L75: Because these specific names haven't been previously mentioned, it is best to write out he genus names
L90: The two commas on this line are not needed
L101: The period before "2)" should be a comma
L116: Missing a tab at the beginning of the paragraph
L139: I believe that "next" could be omitted here
L148: It should be "RAxML" (instead of "RaxML")
L162-163: "Data set using in this analysis was according to data using in gene three analysis" -- I think this would make more sense as "The data set used in this analysis was the same as the data used in the three gene analyses.", but I'm still not sure that I'm following what it is trying to say, so it'd be good to clarify what this statement means.
L163-165: "To annotated" should be "To annotate" and "in the phylogenetic tree" should be "on the phylogenetic tree".
L170,249,295: "BASP" should be "BAPS"
L185-186: "posterior probability density of the combined tree file were..." should be either "posterior probability densities of the combined tree files were..." or "posterior probability density of the combined tree file was..."
L198: "two clades well-support" should be "two well-supported clades"
L273: "This area have" should be "This area has"
L273: This sentence seems to be two that were inadvertently put together, and it seems that the comma following "3 Mya" should be a period.
L339: "share" should be "shares"
L390: "genes tree" should be "gene trees"
L418-419: Should be "between the central basins across several time periods during the Pleistocene"
L441: "low divergences rates" should be "low divergence rates"
Table 2: "BASP" should be "BAPS" x2
Fig. 4: Why are the individuals for Clade II collapsed, while those in Clade I are not?

Reviewer 2 ·

Basic reporting

In this article, Betancourt-Resendes and collaborators perform a phylogeographic analysis of the fish Chirostoma jordani across the Mesa Central of Mexico using mitochondrial and nuclear DNA sequence data. The authors find two phylogenetic groups for the species and speculate about their biogeographical implications.

Overall, the article presents sufficient background and context for the case of study and is well supported by the literature and the extensive experience of the authors. Please, revise the bibliography section for accuracy and consistency in the format used. Also, I found a few references in the text that are missing in the bibliography, but I haven’t carried out and extensive revision of all the references.

The article follows a clear structure and is written in English that will be understandable by most readers. However, in my opinion, the quality of the English could be greatly improved. There are several parts of the manuscript where the writing is cumbersome and hard to follow, including spelling errors and verb tense errors. Some specific edits and comments are made in the section below, but I haven’t carried out an extensive revision of the text.

The figures are correct and convey their purpose of transmitting the main results of the study. I think Figure 2 contains an error and the names of the main clades are flipped. Other suggestions that may improve readability of the figures are suggested below.

The sequence data are uploaded to GenBank but not available yet. It would be nice to see the correspondence between the sample IDs and their accession numbers for each locus. This could be added to Table 1 or in a supplementary table.

Experimental design

This study contributes to the general understanding of the evolution of biological diversity in Central Mexico by showing the intricate phylogeographic patterns that result from the extreme geological complexity of this region.

The study includes extensive sampling across the whole area of distribution of the species, and I consider that the molecular data collected are sufficient to answer the questions proposed.

The analytical methods used are also appropriate but there are parts where the description of the methodology is unclear as well as their interpretation.

Specific comments about the methodology:
- The components of the PCR reactions are not described in the text. The composition of the mastermix could be included in Table S1 together with the thermocycling conditions. Also, the primer sequences could be moved from the main text into the supplementary file.
- L134-136. I suggest to move this part ahead in the section where the other samples used are described. The tags and codes for each sample could also be included in Table 1.
- L149. Do you mean “tree gene partitions” or “THREE gene partitions”?
- L154. Maybe change “To determine the geographic correspondence of haplotypes” to something like “To explore the geographic distribution of haplotypes”?
- L155. Are you sure you used HaploView to construct a haplotype network? It doesn’t seem to me like the right software to do this. Maybe Haploviewer (Salzburger et al. 2011, Molecular Ecology)? Please, double-check and change the text and references if necessary.
- L159-169. This part is confusing. It is not clear to me how you performed these analyses and what data you used. Be clearer about what do you mean by “Data set using in this analysis was according to data using in gene three analysis”. Also, the results could be clearer if you show in the phylogenetic tree the cluster assignment of each sample. You can use colored bullets for the mtDNA and the nDNA results or combine the phylogenetic tree with a chart with two columns where you indicate the BAPS cluster… In my opinion, the written description of these results is too long and complicated to follow for readers that are not familiar with the region. I think that a general written explanation combined with a graphical representation of the results could do a better job.
- L161. References of Lu et al. and Hill et al. are missing in the Bibliography.
- Despite being commonly used, testing the significance of the groupings that result from a clustering analysis like BAPS is not statistically correct. The genetic data used to infer the clusters are the same that the data used to test if the clusters are significantly different, resulting in a case of nonindependence or circularity. It would be OK, however, to report and discuss the PhiST, PhiCT, PhiSC values obtained, but not the P-values, since these are meaningless. See Meirmans (2015) in Molecular Ecology Resources for a great explanation about why this practice is wrong and how to fix it (https://doi.org/10.1111/mec.13243).

Validity of the findings

My major concern about this article is the interpretation of some of the results, specifically those regarding the phylogenetic analysis. I think that it is important that the authors acknowledge that the two clades recovered are not supported by any of the phylogenetic analysis performed. The only node that shows high support is the monophyly of C. jordani but not for each of the clades. I think that if the clades are not supported the discussion about the taxonomy of C. jordani should be revised. For example, I suggest that the authors include a topology test (e.g. AU test?) to assess if these clades are statistically supported by the data, and that they report what is the genetic distance between the two clades. Similarly, I think that the “haplogroups” that the authors claim to result from the haplotype network analyses are highly biased by the phylogenetic results. It is true that there are highly differentiated haplotypes from the rest, but the breaks of the haplogroups are subjective. I think that it is better if the authors indicate the haplotypes that correspond to clades I and II instead of implying that these haplogroups are a result of their analysis. It is only a matter of interpretation but the distinction is important.

About the biogeographical implications, I think that some parts of the discussion are extremely long and detailed, especially those referring to Clade I (L291-400). I can understand the difficulty of correlating the genetic structure results with the extremely complex geological history of the region, but there is no specific biogeographical test or analysis, and therefore the authors can only provide a speculative explanation for the patterns recovered. I think that a more general discussion, that doesn’t address the specific contingencies of each sampling locality, could be more appropriate.

Additional comments

Minor comments and edits (not exhaustive):
- L22: Change “process” to “processes”.
- L27: “species in THE Mexican Plateau”
- L29: Change “has acted to promote” to “has promoted”
- L31: Maybe change to active voice? Something like: “We used mitochondrial (cytb and D-loop) and nuclear (First intron of ribosomal protein S7) data and conduct a phylogeographic and coalescent-based analysis to elucidate the evolutionary history of C. jordani throughout its distribution range in the Mexican Plateau.
- L37-39: Maybe change to : “Clade I occurs exclusively in the north-western basin and shows population structure. Clade II is widespread/widely distributed across the west, central, and eastern basins without population structure.
- L39-42: I think this sentence needs to be clarified. For example: “The split between the main groups was estimated ~1.4 Mya. This cladogenetic event may be associated to the … [break up and clarify]”
- L45: Change “process” to “processes”, “the major goal” to “a/one major goal”.
- L46-50. I wonder if this applies only to riverine species or to all freshwater fishes?
- L51-52: Maybe change to: “a good model system in which to integrate macro and micro-evolutionary studies”
- L56: Maybe change “dynamism of genesis” to “dynamic genesis”
- L57: Maybe change to “ These geological processes are also the main mechanisms that limit or enhance gene flow between populations…”
- L67-73. I think this section could use some rewording and clarification. For example: “The Mesa Silverside is one of the smallest silverside species and occurs in both lentic and lotic ecosystems. It is considered that its ecological versatility is one of the factors behind its wide distributional range (Barbour, 1973a). The species shows low genetic differentiation between some geographically isolated populations (Bloom et al., 2009), but high phenotypic plasticity between lentic and lotic habitats, suggesting that specific environmental forces drive body shape differentiation, mainly in body shape and mouth position (Foster et al., 2015).”
- L80: Maybe change to “Due to the morphological plasticity of this species, DNA sequence data have been useful to test phylogenetic…”
- L86: Change “, and this” to “that”.
- L112: Change “preserve” to “preserved”, and “The fish” to “Whole fish”
- L127: Remove “positive”.
- L129: Change “perform” to “performed”.
- L130: Change to “We translated the cytb alignments to amino acids…”
- L134: Remove “also”.
- L139: Change “to next” to “with”
- L140-142: Maybe substitute “to” between accession numbers by a dash “–”
- L149: “tree” or “three gene partitions”?
- L163: Change to “gene tree analysis”.
- L163-165 and L163-165. Rewrite these sentences.
- L170 and elsewhere: Change “BASP” to “BAPS” across the whole manuscript.
- L179: Change “We estimate” to “We estimated”.
- L181: Change “ParttitionFinder” to “PartitionFinder”.
- L186: Change “were” to “was”
- L204: It is not true that the two clades are well supported. See my comments above. This is a major concern about the interpretation and discussion of the results.
- L224-234. This long list of localities and numbers is a bit confusing for readers that are not familiar with the region. See my comments above regarding clarification of the BAPS results.
- L236. A likelihood value is meaningless on its own, you need a comparison to be able to assess if the value is high, low, better, or worse than others…
- L248-251. See comments above about testing the significance of genetic clusters. Also, it is important to mention that most of the variation still exist among localities rather than among groups.
- L250: Actually, this is not supported in the nDNA data because there is more variation within than among groups (PhiSC > PhiCT).
- L267: Change “date” to “dated”.
- L271: Maybe change to “The biogeographic scenario that explain the cladogenetic events in C. jordani…”
- L273: Change “have” to “has”
- L278-282: Maybe rephrase to something like: “The influence of the geological activity of the “triple junction” in the evolutionary patterns of the central Mexican ichthyofauna has already been noticed.
- L315: It is a bit confusing that suddenly this group is called C. mezquital without previous introduction.
- L328: Remove “than”
- L364: Change “as the cyprinids” to “like the cyprinids”.
- L367: Reference for Beltrán-Álvarez et al. 1997 is missing in the Bibliography.
- L418-430: Aren’t all these arguments also valid to explain patterns in Clade I? What is the difference?
- L431, L449: Change “Clade 2” to “Clade II”.
- L471-477: I think this is last paragraph needs to be rewritten and clarified.
- Figure 1. Maybe it would be helpful to add the numbers of the localities next to biogeographic regions in the legend. There are a lot of colors and some of them are hard to differentiate when they are all mixed up in the map.
- Figure 2. I think that this figure contains an error, and the names of the clades are flipped. Also, maybe this figure could be edited so that the long branches between ingroup and outgroup are shortened because they don’t provide any relevant information, and the height of the figure is increased so the names of the samples can be read more easily. Also, the figure could be more informative if the sample names include a reference to the numbers used in Fig. 1, and as already mentioned above, this figure could include the assignment of each sample to the BAPS clusters. In the legend, change “probability” for “bootstrap”.
- Figure 3. Change “locus” for “loci”.
- Figure 4. Change “Calibrate analysis in C. jordani populations based in” for “CalibrateD analysis OF C. jordani populations based ON”. Add to the legend the meaning of the values above nodes. Also, add “95%” before “HPD”, or the corresponding value.
- References: revise that all the references cited in the text are included here. Check the format and be consistent: capitalization, italics, abbreviations… For example: some references have the full name of the first author, some don’t; some references have periods after the initials, some don’t; L. 506: Change “Talune” for “Tulane”, L. 507: Change “sosa” for “Sosa”; L537: Change “atherinopsidae” for “Atherinopsidae”, L. 553. Remove quotations marks…

---

## Round 0.2 · accepted · Accept

All comments have been addressed and I believe the manuscript is ready for publication. Congratulations!